# Formal Representations of Classical Planning Domains

**Primary Keywords:** *(4) Theory*

## Abstract

Planning domains are an important notion, e.g. when it comes to restricting the input for generalized planning or learning approaches. However, domains as specified in PDDL cannot fully capture the intuitive understanding of a planning domain. We close this semantic gap and propose using PDDL axioms to characterize the (typically infinite) set of legal tasks of a domain. A minor extension makes it possible to express all properties that can be determined in polynomial time. We demonstrate the suitability of the approach on established domains from the International Planning Competition.

## Introduction

Intuitively, a planning domain consists of a typically infinite set of related planning tasks. The concept is important for different areas of classical planning, e.g. for generalized planning, where a single generalized plan should solve all tasks of a domain (Srivastava, Immerman, and Zilberstein 2011; Bonet and Geffner 2018; Francès et al. 2019; Illanes and McIlraith 2019; Drexler, Seipp, and Geffner 2021) or for learning-based approaches to domain-independent planning as in the learning track of the International Planning Competition (IPC) (Fern, Khardon, and Tadepalli 2011).

Planning tasks expressed in the PDDL formalism consist of a "domain description" and a "problem description", and tasks that belong to the same domain share a domain description. However, PDDL domain descriptions only *over-approximate* planning domains: every task that belongs to the same domain can use the same domain description, but not every task that fits a PDDL domain description is part of the (intuitive) domain that we intend to capture. For example, legal tasks in the Blocksworld domain can never include cyclic stacks of blocks (e.g., block $A$ resting on block $B$ resting on block $A$), but PDDL domain descriptions do not include this information. Rather, they are conveyed informally via natural-language descriptions of planning domains (Helmert 2003; Hoffmann 2005; Drexler, Seipp, and Geffner 2021) or implicitly by unspoken convention.

Where a formal notion is necessary, for example where domains are inputs to an algorithm, domains are sometimes defined as *finite* sets of tasks (Lotinac et al. 2016; Segovia, Jiménez, and Jonsson 2016; Jiménez, Segovia-Aguas, and Jonsson 2019), but this is only suitable for certain contexts. Theoretical results trivialize in this finite setting, and it is unsuitable for stating algorithmic problems such as generating additional example tasks in a given domain. Closing the semantic gap between a PDDL domain and the intuitive notion of a domain with an exact formal characterization of all tasks in the domain would open up many opportunities.

For example, we can envision a domain-independent instance generator that algorithmically creates random tasks in an arbitrary planning domain given as input. This would enable more diverse benchmarks, which is not only useful for comparing planning systems but also for improving their robustness towards unexpected corner cases. Such a task generator would also be highly useful for many learning-based approaches that rely on a high number of sample tasks.

The additional domain knowledge of the formal characterization makes it possible to extract and prove invariants such as mutexes already on the domain level. Such invariants would also be highly interesting for generalized planning, where it might even be feasible to automatically verify correctness of a generalized plan across an entire domain.

Formally defined planning domains might also enable certain forms of model checking, answering questions like whether the domain has unsolvable tasks or whether there can be states with certain properties.

We propose a formalism for such a formal characterization of planning domains that is based on PDDL axioms. Such axioms are an established concept in planning and a powerful language feature whose usefulness has already been proven theoretically (Thiébaux, Hoffmann, and Nebel 2005) and practically (Ivankovic and Haslum 2015).

We first present the necessary background on classical planning and Datalog with negation. We then introduce and discuss a first formalism for characterizing planning domains. A subsequent extension allows us to express all properties that can be decided in polynomial time. We conclude with examples on established IPC domains that showcase the suitability of our approach.

## Background

We introduce the necessary background from classical planning and the logical query language Datalog$^\neg$, which can be seen as a counterpart of PDDL axioms.

### PDDL Axioms, Domains, and Tasks

We assume that the reader is familiar with first-order logic.

In PDDL, the task specification is separated into a *PDDL domain* and a *PDDL problem*. The PDDL domain specifies (parts of) the language as well as the dynamics of the task, i. e., how the world states can get altered. The PDDL problem defines additional constants, the initial state and the goal. PDDL also supports typing (as in many-sorted logic), which we omit to simplify the presentation.

The dynamics of the tasks are given by operators and axioms. Operators have a *precondition* that describes when they are applicable and an *effect* that describes how they affect the current world state.[1] In contrast to operators, whose application can be controlled, axioms *must* be evaluated after every operator application. We follow the definition of PDDL axioms by Thiébaux, Hoffmann, and Nebel (2005):

**Definition 1** ((Stratifiable) PDDL axioms). *A PDDL axiom is a pair $\langle \phi, \psi \rangle$ such that $\phi$ is a first-order atom and $\psi$ is a first-order formula, where $\psi$ and $\phi$ have the same set of free variables. We write the axiom $\langle \phi, \psi \rangle$ as $\phi \leftarrow \psi$ and call $\phi$ the* head *and $\psi$ the* body *of the axiom.*

*A set $\mathcal{A}$ of PDDL axioms is stratifiable iff there exists a partition (stratification) $\mathcal{P}_1, \ldots, \mathcal{P}_n$ of the predicates such that for every $P_i \in \mathcal{P}_i$ and $P_i(\bar{x}) \leftarrow \psi(\bar{x}) \in \mathcal{A}$*

- *if $P_j \in \mathcal{P}_j$ appears in $\psi(\bar{x})$ then $j \leq i$, and*
- *if $P_j \in \mathcal{P}_j$ appears negated in the translation of $\psi(\bar{x})$ to negation normal form then $j < i$.*

The predicates can be distinguished into basic and derived predicates, where the derived predicates are the ones occurring in the head of an axiom. Operator effects may not (directly) modify derived predicates. For sets $\mathcal{C}$ of constants and $\mathcal{P}$ of predicates, we write $\Sigma_{\mathcal{C},\mathcal{P}}$ to refer to the (function-free) first-order signature $(\mathcal{C}, \mathcal{P})$.

A *state* for $\Sigma_{\mathcal{C},\mathcal{P}}$ is a Herbrand interpretation and can thus also be specified as a truth assignment for the ground atoms over the predicates and constants. A *basic state* only specifies the truth of the ground atoms over basic predicates: if $\mathcal{P}_b \subseteq \mathcal{P}$ are the basic predicates, then a *basic state* for $\Sigma_{\mathcal{C},\mathcal{P}}$ is a state for $\Sigma_{\mathcal{C},\mathcal{P}_b}$.

The axioms extend basic state $s$ to the *extended state* $[\![s]\!]$. For a set $\mathcal{A}$ of axioms, a stratification $\mathcal{P}_1, \ldots, \mathcal{P}_n$ induces a partition $\mathcal{A}_1, \ldots, \mathcal{A}_n$ as $\mathcal{A}_i = \{P_i(\bar{x}) \leftarrow \psi(\bar{x}) \in \mathcal{A} \mid P_i \in \mathcal{P}_i\}$. The extended state can be computed with a sequence of least fixed point computations induced by this axiom partition (Algorithm 1). It maintains a state. Initially only the atoms that are true in basic state $s$ are true. For every stratum, it then successively "applies" instantiations of the axioms in the stratum until a fixpoint is reached, making the head of rules true for which the body is already true in the current state. The result is independent of the stratification and the order in which axioms within each block are considered (Apt, Blair, and Walker 1988; Thiébaux, Hoffmann, and Nebel 2005). Stratifications can be determined efficiently (Thiébaux, Hoffmann, and Nebel 2005).

A PDDL task is given by a PDDL domain and a PDDL problem for the domain with the following components:

---

[1]The details of the operator specification are not relevant for this work.

---

**Algorithm 1:** Extension of a basic state

**function** EXTEND-STATE(Axiom partition $\mathcal{A}_1, \ldots, \mathcal{A}_n$, constants $\mathcal{C}$, basic state $s$)

$s' :=$ truth assignment to all ground atoms with
$$s'(a) := \begin{cases} s(a) & \text{if the predicate of } a \text{ is basic} \\ false & \text{if the predicate of } a \text{ is derived} \end{cases}$$

**for** $i \in \{1, \ldots, n\}$ **do**

   **while** there exists a rule $\phi \leftarrow \psi \in A_i$ and a substitution $\sigma$ of the free variables of $\psi$ with constants such that $s' \models \psi\{\sigma\} \wedge \neg\phi\{\sigma\}$ **do**

      Choose such a $\phi \leftarrow \psi$ and $\sigma$.
      $s'(\phi\{\sigma\}) := true$

**return** $s'$

---

**Definition 2** (PDDL domain). *A PDDL domain is a tuple $\langle \mathcal{P}, \mathcal{C}, \mathcal{O}, \mathcal{A} \rangle$, where $\mathcal{P}$ is a finite set of predicate symbols, $\mathcal{C}$ is a finite set of constant symbols, $\mathcal{O}$ is a finite set of PDDL operators over $\Sigma_{\mathcal{C},\mathcal{P}}$ and $\mathcal{A}$ is a stratifiable finite set of PDDL axioms over $\Sigma_{\mathcal{C},\mathcal{P}}$.*

**Definition 3** (PDDL problem). *A PDDL problem is a tuple $\langle \mathfrak{D}, \mathcal{C}, \mathcal{I}, \mathcal{G} \rangle$, where $\mathfrak{D} = \langle \mathcal{P}, \mathcal{C}_{\mathfrak{D}}, \mathcal{O}, \mathcal{A} \rangle$ is a PDDL domain, $\mathcal{C}$ is a finite set of constant symbols disjoint from $\mathcal{C}_{\mathfrak{D}}$, $\mathcal{I}$ is a basic state (the initial state) and $\mathcal{G}$ a first-order sentence over $\Sigma_{\mathcal{C} \cup \mathcal{C}_{\mathfrak{D}}, \mathcal{P}}$.*

The aim is to find a sequence of operators that leads from the initial state to one satisfying the goal. Operator preconditions and the goal are evaluated on extended states, so the initial state and the basic state after each operator application must be extended to interpret the derived predicates.

### Datalog with Negation

The semantics of PDDL axioms has a strong correspondence to stratified Datalog¬. Datalog typically uses a slightly different notation for rules and the terminology from database theory speaks of *extensional* instead of basic predicates and *intensional* instead of derived predicates. Here we will present the concepts using the terminology and notation we used for PDDL axioms. Indeed, we can specify stratified Datalog¬ as a syntactic restriction on PDDL axioms.

**Definition 4.** *A Datalog¬ rule is a PDDL axiom $\phi(\bar{x}) \leftarrow \exists \bar{y}\, \psi(\bar{x}, \bar{y})$, where $\bar{x}$ are the variables mentioned in the head, and the body $\psi$ is a conjunction of literals that mentions exactly the variables from $\bar{x}$ and $\bar{y}$.*

A set of Datalog¬ rules is *semipositive* if no derived predicate occurs negated in the body of any rule.

Interestingly, first-order queries can be rewritten in linear space into nonrecursive Datalog with negation (Abiteboul, Hull, and Vianu 1995), which is a subset of stratified Datalog¬. Thus we can rewrite every stratifiable set of PDDL axioms into a stratifiable set of Datalog¬ rules (Thiébaux, Hoffmann, and Nebel 2005).

## Formal Characterization of Planning Domains

Our aim is to introduce a formalism that enables us to exactly express which tasks belong to a certain planning do-

main. A planning domain is a possibly infinite set of planning tasks that share some characteristics that make them intuitively "belong together", but it depends on the application and context what this means exactly. Thus, the formalism needs to be expressive enough to be useful for a wide range of such intuitions but at the same time it should be sufficiently restrictive to meaningfully support it in domain-independent algorithms, e.g. for learning, generalized planning or task generation. We also require that the specification of the legal tasks can be concisely expressed and that we can efficiently test whether a given task belongs to the domain.

For practical reasons, we also aim for a definition that is compatible with existing PDDL domains, which specify the dynamics of the domain but do not restrict the tasks beyond the permitted predicates. This will allow us to augment existing benchmarks with an exact characterization.

In the following, we propose a first formalism based on axioms and will argue that it already satisfies most of these requirements. Later, we will add a minor extension ($<$) known from database theory that allows us to specify any property that can be tested in polynomial time.

For our definition, we build on the notion of a *query* from database theory and apply it to PDDL axioms. For our purpose we only need queries without output parameters (0-ary queries):

**Definition 5** (Query). *Let $\mathcal{A}$ be a finite stratifiable set of PDDL axioms and let $P$ be a $0$-ary predicate symbol. The query $Q_{\mathcal{A},P}$ is the function*

$$Q_{\mathcal{A},P}(\mathcal{C}, s) = \begin{cases} \top & \text{if } [\![s]\!] \models P() \\ \bot & \text{otherwise}, \end{cases}$$

*where $\mathcal{C}$ is any set of constants, $s$ is a basic state over the predicates in $\mathcal{A}$ and constants from $\mathcal{C}$, and $s$ is extended wrt. the axiom set $\mathcal{A}$ and constant set $\mathcal{C}$.*

Note that the query is not the *result* of evaluating the truth of the predicate in the extension of a given state but a *mapping* from basic states to the corresponding result. For simplifying the presentation, we will mostly use the same predicate *legal* for the query predicate, but this is not a requirement of the definition:

**Definition 6** (Planning domain (without order)). *A planning domain is a tuple $\langle \mathcal{P}, \mathcal{C}, \mathcal{O}, \mathcal{A}, L, \mathcal{G} \rangle$, where*

- *$\mathcal{P}$ is a finite set of predicate symbols that can be partitioned into two sets, the set of basic predicates $\mathcal{B}$ and the set of derived predicates $\mathcal{D}$,*
- *$\mathcal{C}$ is a finite set of constant symbols,*
- *$\mathcal{O}$ is a finite set of PDDL operators over $\Sigma_{\mathcal{C},\mathcal{P}}$,*
- *$\mathcal{A}$ is a stratifiable finite set of PDDL axioms over $\Sigma_{\mathcal{C},\mathcal{P}}$.*
- *$L$ is the $0$-ary query predicate with $L \in \mathcal{D}$, and*
- *$\mathcal{G}$ is a first-order sentence over $\Sigma_{\mathcal{C},\mathcal{P}}$.*

The major differences to PDDL domains are the existence of the query predicate and the domain-wide specification of the goal. We will later discuss this decision. Although the definition of a planning domain and a PDDL domain are very similar, in practice it will not be just an existing PDDL domain plus $L$ and $\mathcal{G}$, but the axioms and derived predicates will contain additional elements that are not relevant for the operators but are important for deciding the legality of a task.

Since the goal has already been fixed, a specific problem only specifies additional objects and the initial state. It is legal if the query predicate can be derived from the initial state by means of the axioms.

**Definition 7** (Legal Problem (without order)). *A problem for a planning domain $\mathbf{D} = \langle \mathcal{P}, \mathcal{C}, \mathcal{O}, \mathcal{A}, L, \mathcal{G} \rangle$ is a tuple $\langle \mathcal{C}', \mathcal{I} \rangle$, where $\mathcal{C}'$ is a finite set of constant symbols disjoint from $\mathcal{C}$ and the initial state $\mathcal{I}$ is a basic state over $\Sigma_{\mathcal{C} \cup \mathcal{C}', \mathcal{P}}$. The problem is legal for $\mathbf{D}$ if $Q_{\mathcal{A},L}(\mathcal{C} \cup \mathcal{C}', \mathcal{I}) = \top$.*

For the representation size $\|\mathbf{P}\|$ of problem $\mathbf{P}$ we assume that the initial state is specified by an explicit representation of all true ground atoms (as it is the case for states in PDDL). Then testing legality is efficient for a fixed planning domain.

**Theorem 1.** *Let $\mathbf{D}$ be a fixed planning domain. For a given problem $\mathbf{P}$ for $\mathbf{D}$ we can decide whether $\mathbf{P}$ is legal for $\mathbf{D}$ in polynomial time in $\|\mathbf{P}\|$.*

*Proof.* Helmert (2009) describes how PDDL axioms can efficiently be rewritten as stratified Datalog$^\neg$ rules, introducing derived predicates for some subformulas of the axiom body. This is independent of a specific state. Stratified Datalog$^\neg$ is *data-complete* for $\mathsf{P}$ (Apt, Blair, and Walker 1988; Dantsin et al. 2001). This means that for a fixed set of rules and query, the result of the query for a specific basic state (an input database in the database context) can be determined in polynomial time in the size of the state. $\square$

We relate problems for planning domains to the known concept of PDDL problems in the obvious way:

**Definition 8** (Induced PDDL problem). *A problem $\langle \mathcal{C}', \mathcal{I} \rangle$ for planning domain $\langle \mathcal{P}, \mathcal{C}, \mathcal{O}, \mathcal{A}, L, \mathcal{G} \rangle$ induces the PDDL problem $\langle \langle \mathcal{P}, \mathcal{C}, \mathcal{O}, \mathcal{A} \rangle, \mathcal{C}', \mathcal{I}, \mathcal{G} \rangle$.*

Note that in the induced task we did not remove the predicates and axioms that are only relevant for the query. In practice, we would either separate them in the definition of the planning domain or filter them by means of a simple backwards reachability analysis from the predicates mentioned in operators and the goal on the syntactic level of the axioms.

We can now define the typically infinite set of PDDL problems that is characterized by a planning domain.

**Definition 9** (PDDL problems of a Planning domain). *A planning domain $\mathbf{D}$ characterizes the set of all PDDL problems that are induced by some legal problem for $\mathbf{D}$.*

On the level of an individual planning task, it is irrelevant how we distribute the different components to the problem and domain specification, but for characterizing a domain, it becomes a relevant aspect. Fixing the set of (schematic) operators on the domain level is in line with the definition of PDDL domains. Fixing also the goal formula deviates from the established standard in PDDL, where the goal is only a part of the problem specification on top of the domain.

## Goal Specification

In the following, we will argue that our definition of the goal specification is powerful enough to express existing benchmarks domains and that leaving the entire goal specification

to the task would require an additional undesirable mechanism to establish relevant restrictions.

In some IPC benchmark domains, all problems already share the same first-order goal specification. For example, in the full-ADL variant of Miconic (Koehler and Schuster 2000), where a number of passengers needs to be brought to their destination floors with an elevator, there is a joint goal $\forall p(passenger(p) \rightarrow served(p))$. As not all planning systems support the full ADL fragment, IPC domains are often expressed in the STRIPS fragment of PDDL, which requires the goal to be a conjunction of atoms, and thus goals differ between problems. In many cases, the STRIPS goal is already the result of a compilation from a common first-order goal, or the domain designer has an informal common goal in mind (such as "deliver all packages"). With our proposed formalism it is not necessary to reconstruct a common first-order formula but instead we can directly cover any conjunction of ground atoms for the individual problems.

Consider for example the Logistics domain from IPC 1998 (McDermott 2000), where the task is to transport packages to their destination location. Within cities, trucks can transport the packages between locations, and airplanes can transport packages between the airports of different cities. Predicates $at(p,l)$ and $in(p,v)$ express that a package/truck is at a location or a package is in a vehicle. An example goal of such an IPC problem is $at(p_1, l_1) \wedge at(p_2, l_1) \wedge at(p_3, l_3)$. We can cover any such conjunction by means of a new binary predicate $at^g$ and a first-order goal $\forall p, l(at^g(p,l) \rightarrow at(p,l))$, moving the exact requirements of the problem into the initial state specification. In the example, we would extend the original initial state to interpret $at^g$ as $\{(p_1, l_1), (p_2, l_2), (p_3, l_1)\}$.

This way, we can on the problem side support arbitrary conjunctions of ground atoms as goals. Let $\mathcal{P} = \{P_1, \ldots, P_n\}$ be the set of predicates we want to support. Then we would add a new predicate $P^g$ for every $P \in \mathcal{P}$ with the same arity as $P$ and use the goal formula $\bigwedge_{i=1,\ldots,n} \forall \bar{x}_i(P^g(\bar{x}_i) \rightarrow P(\bar{x}_i))$, where $\bar{x}_i$ is a vector of variables with the arity of $P_i$.

We can extend the idea to conjunctions of *literals* by introducing additional goal predicates for the negative literals.

Since the truth of the $P^g$ atoms is defined in the initial state, they can also be analyzed by the query for the legality of the task. Consider as an example a variant of the Logistics domain where trucks can only travel on a road network (encoded by edges $E$ between locations) and there are also goal locations specified for a subset of the trucks. As PDDL axioms can compute transitive closures, we can require that every truck can reach its goal location:

$$reach(t,l) \leftarrow at(t,l) \vee \exists l'(reach(t,l') \wedge E(l',l))$$
$$illegal() \leftarrow \exists t, l(truck(t) \wedge at^g(t,l) \wedge \neg reach(t,l))$$
$$legal() \leftarrow \neg illegal()$$

We have seen that with a joint goal formula, we can still leave some aspects of the goal to the individual problems.

For many applications, e.g. generalized planning where the same general plan must work for all problems of the domain, it is essential that the domain specification can enforce some uniformity of the goals. So if we did not use a common goal, we would need some other mechanism for constraining the legal goals. One could consider to specify a first-order formula $\phi$ in the domain to constrain the problem-specific goal $\gamma$ but it is not even clear what this constraint could look like. Requiring that $\gamma \models \phi$ means that in every task we need to achieve $\phi$ but there can be arbitrary additional requirement. Requiring $\phi \models \gamma$ means that achieving $\phi$ is definitively sufficient but on a domain-specific level there is no information what kind of relaxation $\gamma$ permits, so it is unclear how this can be exploited. Since deciding such logical consequences is a hard problem even for a fixed planning domain, we did not further pursue these ideas.

## Adding Order

In general, semipositive Datalog$^\neg$ is strictly less expressive than stratified Datalog$^\neg$, which in turn is strictly less expressive than fixpoint queries (Abiteboul, Hull, and Vianu 1995). Interestingly, on certain finite structures the difference disappears: If there are predicates $succ(x.y)$, $min(x)$ and $max(x)$ that are always interpreted such that $succ$ is a successor relation of a linear order on the finite universe, and $min$ and $max$ determine the minimal and maximal element in this order then semipositive Datalog$^\neg$ is equivalent to fixpoint queries and both capture P (Papadimitriou 1985; Abiteboul, Hull, and Vianu 1995). The same is true for stratified Datalog$^\neg$, where we do not even need $min$ and $max$ as basic predicates (Abiteboul, Hull, and Vianu 1995).[2]

Let us briefly clarify what it means in our context that a formalism captures P. The following definition is based on the general definition by Libkin (2004) for arbitrary complexity classes, logics and classes of finite structures.

**Definition 10.** *Let $L$ be a logic query language. Then $L$ captures P if both of the following hold:*

- *For a fixed set of rules and query predicate in $L$, testing whether the query is true for a finite structure $\mathcal{S}$ is possible in polynomial time in the size of $\mathcal{S}$.*
- *For every property $P$ of finite structures that can be decided in polynomial time, there is a fixed set of rules and query predicate such that for every finite structure $\mathcal{S}$ the query is true iff $\mathcal{S}$ has property $P$.*

Applied to our context, the finite structures are the basic states and the first condition expresses that we can decide in polynomial time whether the given problem is legal. As we have discussed before, this is already the case for the considered formalisms if we do not require order. The second property is much more interesting: *Every* property of basic states that can (by any mechanism) be tested in polynomial time, can be phrased as a query in stratified Datalog$^\neg$ if there is a successor relation!

Since this is clearly a desirable property for formal representations of planning domains, we will in the following extend the previous definition accordingly.

We do not want to require a successor relation to be explicitly defined in the initial state but instead the result should be equivalent for arbitrary such orders.

_______________

[2]We can derive them from $succ$, e.g. with $min(x) \leftarrow \neg\exists y succ(y,x)$.

For this purpose, we will borrow another concept from Datalog (with or without negation), namely the concept of an *order-invariant* query (Rudolph and Thomazo 2016).

We first define (slightly abusing notation) the union of states over the same set of constants but disjoint predicates.

**Definition 11** (Union of states). *Let $s$ be a state over $\Sigma_{\mathcal{C},\mathcal{P}}$ and $s'$ be a state over $\Sigma_{\mathcal{C},\mathcal{P}'}$, where $\mathcal{P} \cap \mathcal{P}' = \emptyset$. State $s \cup s'$ is the state over $\Sigma_{\mathcal{C},\mathcal{P} \cup \mathcal{P}'}$ that interprets all predicates from $\mathcal{P}$ like $s$ and all predicates from $\mathcal{P}'$ like $s'$.*

A query that uses order is order-invariant if the result is independent of the chosen order.

**Definition 12** (Order-invariant query). *Let $\mathcal{A}$ be a finite set of stratifiable PDDL axioms with basic predicates $\mathcal{B}$ and derived predicates $\mathcal{D}$ and let $P \in \mathcal{D}$ be a 0-ary predicate. Let further succ be a binary predicate in $\mathcal{B}$.*

*Query $Q_{\mathcal{A},P}$ is order-invariant wrt. succ if for every finite set $\mathcal{C}$ of constants and state $s$ over $\Sigma_{\mathcal{C},\mathcal{B}\setminus\{succ\}}$ it holds that if $s_{succ}$ and $s'_{succ}$ are states over $\Sigma_{\mathcal{C},\{succ\}}$ that define a linear order for $\mathcal{C}$ then $Q_{\mathcal{A},P}(\mathcal{C}, s \cup s_{succ}) = Q_{\mathcal{A},P}(\mathcal{C}, s \cup s'_{succ})$.*

A planning domain with order is defined analogously to our previous notion of planning domains, only that there is now a binary basic predicate *succ* that may be used in the axioms for testing legality but not be relevant for the semantics of the induced problems. We thus need to adapt our earlier definition. In particular, we now need to be more precise about what axioms can be used for which purpose. For clarity, we also partition the set of predicates accordingly. Subscript t refers to components that may be used everywhere, while subscript q refers to components that are only used for the legality query.

**Definition 13** (Order-supported planning domain). *An order-supported planning domain is a tuple $\langle \mathcal{P}_t, \mathcal{P}_q, \mathcal{C}, \mathcal{O}, \mathcal{A}_t, \mathcal{A}_q, L, succ, \mathcal{G} \rangle$, where*

- *$\mathcal{P}_t$ is a finite set of predicate symbols that can be partitioned into a set of basic predicates $\mathcal{B}_t$ and a set of derived predicates $\mathcal{D}_t$,*
- *$\mathcal{P}_q$ with $\mathcal{P}_q \cap \mathcal{P}_t = \emptyset$ is a finite set of predicate symbols that can be partitioned into a set of derived predicates $\mathcal{D}_q$ and $\{succ\}$,*
- *$\mathcal{C}$ is a finite set of constant symbols,*
- *$\mathcal{O}$ is a finite set of PDDL operators over $\Sigma_{\mathcal{C},\mathcal{P}_t}$,*
- *$\mathcal{A}_t$ is a stratifiable finite set of PDDL axioms over $\Sigma_{\mathcal{C},\mathcal{P}_t}$.*
- *$\mathcal{A}_q$ is a stratifiable finite set of PDDL axioms over $\Sigma_{\mathcal{C},\mathcal{P}_t \cup \mathcal{P}_q}$.*
- *$L$ is the 0-ary query predicate with $L \in \mathcal{D}_q$,*
- *succ is a binary basic successor predicate,*
- *$Q_{\mathcal{A}_t \cup \mathcal{A}_q, L}$ is order-invariant wrt. succ, and*
- *$\mathcal{G}$ is a first-order sentence over $\Sigma_{\mathcal{C},\mathcal{P}_t}$.*

We need to adapt the definition of legal problems in the obvious way: The initial state only interprets the predicates from $\mathcal{B}_t$ and the legality test uses all axioms.

**Definition 14** (Legal Problem (with order)). *A problem for an order-supported planning domain $\mathbf{D} = \langle \mathcal{P}_t, \mathcal{P}_q, \mathcal{C}, \mathcal{O}, \mathcal{A}_t, \mathcal{A}_q, L, succ, \mathcal{G} \rangle$ is a tuple $\mathbf{P} = \langle \mathcal{C}', \mathcal{I} \rangle$, where $\mathcal{C}'$ is a finite set of constant symbols disjoint from $\mathcal{C}$ and the initial state $\mathcal{I}$ is a basic state over $\Sigma_{\mathcal{C} \cup \mathcal{C}',\mathcal{P}_t}$.*

$\mathbf{P}$ *is legal for $\mathbf{D}$ if $Q_{\mathcal{A}_t \cup \mathcal{A}_q, L}(\mathcal{C} \cup \mathcal{C}', \mathcal{I} \cup s_{succ}) = \top$ for some (or equivalently every) state $s_{succ}$ over $\Sigma_{\mathcal{C},\{succ\}}$ that defines a linear order for $\mathcal{C}$.*

The requirement that there must be an interpretation of *succ* such that the query is true is easy to test in practice: since the query is order-invariant, all orders will lead to the same result and we can use an arbitrary order of the objects to define $s_{succ}$.

A natural question is, whether requiring order invariance is harmful for the theoretical properties of the approach, which is not the case:

**Theorem 2.** *Order-invariant PDDL axiom queries capture* P.

*Proof sketch.* The result that semipositive Datalog$^\neg$ on ordered databases with *min* and *max* can express all P queries is based on its equivalence to fixpoint queries. The corresponding proof creates for a fixpoint query an equivalent *order-invariant* semipositive Datalog$^\neg$ program that uses the order to "iterate" over all objects in the universe (Abiteboul, Hull, and Vianu 1995, Lemma 15.4.7). □

The last definition we need to adapt is the one for the induced PDDL problem, that does not include the predicates and axioms for the legality test:

**Definition 15** (Induced PDDL problem (with order)). *Problem $\langle \mathcal{C}', \mathcal{I} \rangle$ for order-supported planning domain $\langle \mathcal{P}_t, \mathcal{P}_q, \mathcal{C}, \mathcal{O}, \mathcal{A}_t, \mathcal{A}_q, L, succ, \mathcal{G} \rangle$ induces the PDDL problem $\langle \langle \mathcal{P}_t, \mathcal{C}, \mathcal{O}, \mathcal{A}_t \rangle, \mathcal{C}', \mathcal{I}, \mathcal{G} \rangle$.*

In a later example we exploit order support to verify that a relation encoding edges defines a square grid graph.

## Example Domains

We now provide examples on how our approach can be useful to constrain legal tasks beyond the possibilities of a PDDL domain. In all examples we use the original PDDL domain and initial states from the IPC and only move some information from the problem goals into the initial state.

### Blocksworld

As a first example, we consider the well-studied Blocksworld domain (Gupta and Nau 1992; Slaney and Thiébaux 2001). In contrast to an earlier logic axiomatization (Cook and Liu 2003), we use the variant and encoding used in the IPC 2000 (Bacchus 2001). The objects in Blocksworld are blocks. Atom $on(a, b)$ encodes that block $a$ is on top of block $b$ and $ontable(a)$ that block $a$ is on the table. Atom $clear(b)$ indicates that no block lies on $b$. There is a hand that can be used to move blocks around, where $holding(b)$ expresses that the hand holds $b$ and $handempty()$ that it does not hold any block. The goal of all Blocksworld tasks from the IPC is to stack the blocks into a single, fully specified tower.

The initial state always describes a physically meaningful configuration of the blocks with an empty hand. With the given predicates, we can easily specify impossible configurations, e.g. where block $b$ is on itself, block $a$ is on $c$ and

$c$ on $a$, or where $d$ is at the same time on the table and on several other blocks.

In all examples we will use *legal* as the query predicate but the axioms will first derive predicate *illegal* and establish *legal* with the axiom $legal() \leftarrow \neg illegal()$.

The following axioms are based on the ones by Cook and Liu (2003) for a Blocksworld encoding without a hand and a single predicate *above*.

We define the derived predicate *above* to be the transitive closure of *on* and use it to forbid cyclic towers, i.e. towers where a block is above itself.

$$above(x, y) \leftarrow on(x, y) \lor \exists z \, (on(x, z) \land above(z, y))$$
$$illegal() \leftarrow \exists x \, above(x, x)$$

In the Blocksworld domain blocks can only be stacked to build simple towers, i.e. there is at most one block under each block and there is at most one block on each block.

$$illegal() \leftarrow \exists x, y, z \, (on(x, y) \land on(x, z) \land y \neq z)$$
$$illegal() \leftarrow \exists x, y, z \, (on(y, x) \land on(z, x) \land y \neq z)$$

In the initial state no block may be held.

$$illegal() \leftarrow \neg handempty() \lor \exists x \, holding(x)$$

Furthermore, the position of each block must be unambiguously identified. With the previous two rules preventing blocks to be in the hand, we can define the position of each block to be unique with the following axioms.

$$illegal() \leftarrow \neg \forall x (ontable(x) \lor \exists y \, on(x, y))$$
$$illegal() \leftarrow \exists x (ontable(x) \land \exists y \, on(x, y))$$
$$illegal() \leftarrow \neg \forall x (clear(x) \lor \exists y \, on(y, x))$$
$$illegal() \leftarrow \exists x (clear(x) \land \exists y \, on(y, x))$$

The IPC problem goals are conjunctions of atoms $on(x, y)$ that define a single tower. To encode them, we use a new predicate $on^g$ and the domain-wide goal $\forall x, y(on^g(x, y) \rightarrow on(x, y))$. To require a single tower, we use analogous axioms as above to ensure simple, non-cyclic towers (with $on^g$ instead of *on*, and new derived predicate $above^g$). We identify the bottom block, make sure that there is exactly one such block and require that all other blocks are above the bottom block:

$$bot(x) \leftarrow \neg \exists y \, on^g(x, y)$$
$$illegal() \leftarrow \exists x, x'(x \neq x' \land bot(x) \land bot(x'))$$
$$illegal() \leftarrow \neg \exists x \, bot(x)$$
$$illegal() \leftarrow \exists x(bot(x) \land \exists x'(x \neq x' \land \neg above^g(x', x)))$$

### Floortile

In the Floortile domain (Linares López, Celorrio, and Olaya 2015) there is a number of robots that should paint the tiles in a rectangular grid. The domain uses unary predicates *tile*, *robot* and *color* to distinguish the objects into three types. Binary predicates *up*, *down*, *right* and *left* specify the relative location of adjacent tiles. Moreover, there are predicates $robot\text{-}at(r, t)$ to express that robot $r$ is on tile $t$ and $painted(t, c)$ to specify that tile $t$ has been colored with color

$c$. Atom $clear(t)$ indicates that there is no robot on tile $t$ and that $t$ is not colored.

Drexler, Seipp, and Geffner (2021) require for the domain that there is at most one robot on each tile[3] and otherwise the tile is clear. Initially the tiles are unpainted and "the goal is to paint a rectangular subset of the grid in chessboard style.". The goal in the PDDL problem specifies for a subset of the tiles the exact color they should be painted.

There are two very interesting aspects in this domain that we will discuss: the requirement that predicates $up, down, \ldots$ encode a rectangular grid and that the goal encode a chessboard-kind pattern.

In PDDL, typing already enforces that the parameters of each true predicate are of the correct types. Since we did not introduce typing, we show exemplarily how we can ensure that the first parameter of $robot\text{-}at$ is a robot and the second one a tile, and in the following assume that there are analogue axioms for the other predicates as well:

$$illegal() \leftarrow \exists r, t(robot\text{-}at(r, t) \land (\neg robot(r) \lor \neg tile(t)))$$

The following axioms enforce that there is at most one robot on each tile and that a tile is clear iff it is not colored and there is no robot on it:

$$illegal() \leftarrow \exists t, r, r'(r \neq r' \land robot\text{-}at(r, t) \land robot\text{-}at(r', t))$$
$$illegal() \leftarrow \exists t(clear(t) \land$$
$$(\exists r \, robot\text{-}at(r, t)) \lor \exists c \, painted(t, c))$$
$$illegal() \leftarrow \exists t(tile(t) \land \neg clear(t) \land \neg \exists r \, robot\text{-}at(r, t) \land$$
$$\neg \exists c \, painted(t, c))$$

**Rectangular Grid** We ensure that *up* is the inverse of *down* and both are irreflexive (analogously for *left* and *right*). We then make sure that every tile has at most one adjacent tile in every cardinal direction (exemplarily shown for *left*).

$$illegal() \leftarrow \exists t, t'(up(t, t') \land \neg down(t', t))$$
$$illegal() \leftarrow \exists t, t'(down(t, t') \land \neg up(t', t))$$
$$illegal() \leftarrow \exists t \, up(t, t)$$
$$illegal() \leftarrow \exists t \, down(t, t)$$
$$illegal() \leftarrow \exists t, t', t''(left(t, t') \land left(t, t'') \land t' \neq t'')$$

The verification that the predicates encode a grid is based on a bijection that assigns every tile a coordinate in the form of a row and a column. The challenge is that we cannot use additional objects to represent rows and columns. Instead we will use the tiles on the top fringe to represent the corresponding columns and those on the left fringe for the rows.

The top left corner will thus identify the first column and row. We first identify all tiles that have no tile to the left or top and verify that there is only one such top left (*TL*) tile. We can analogously verify that there is exactly one tile for the other three corners.

$$TL(t) \leftarrow \neg \exists t'(left(t, t') \lor up(t, t'))$$
$$illegal() \leftarrow \exists t, t'(TL(t) \land TL(t') \land t \neq t')$$

---

[3]They do not require that every robot is on exactly one tile but this would be equally easy to formulate.

From the top left tile, we identify all tiles that are right of it as column-defining (*col*) tiles (and all tiles below it analogously as *row* tiles).

$$col(t) \leftarrow TL(t)$$
$$col(t) \leftarrow \exists t'(col(t') \wedge right(t', t))$$

Predicate $colOf(t, c)$ represents that tile $t$ is in column $c$ (predicate *rowOf* analogously for rows):

$$colOf(t, t) \leftarrow col(t)$$
$$colOf(t, c) \leftarrow \exists t'(colOf(t', c) \wedge down(t', t))$$

We will now ensure that the mapping from row/column pairs to vertices is a bijection.

The mapping induced by *colOf* and *rowOf* is a function if for every row/column pair there is exactly one tile.

$$illegal() \leftarrow \exists r, c(row(r) \wedge col(c) \wedge$$
$$\neg \exists t(rowOf(t, r) \wedge colOf(t, c)))$$
$$illegal() \leftarrow \exists r, c, t, t'(t \neq t' \wedge$$
$$rowOf(t, r) \wedge colOf(t, r) \wedge$$
$$rowOf(t', r) \wedge colOf(t', r))$$

It is surjective if for every tile there is a row and column.

$$illegal() \leftarrow \exists t(\neg \exists r \, rowOf(t, r) \vee \neg \exists c \, colOf(t, c))$$

It is injective if no tile is in two columns or rows.

$$illegal() \leftarrow \exists r, r', t(r \neq r' \wedge rowOf(t, r) \wedge rowOf(t, r'))$$
$$illegal() \leftarrow \exists c, c', t(c \neq c' \wedge colOf(t, c) \wedge colOf(t, c'))$$

Since columns were purely based on *right* and rows on *down*, we still need to ensure that these describe a grid. We can do this by verifying that moving right and down leads to the same tile as moving down and right and vice versa (analogously for other directions).

$$illegal() \leftarrow \exists t, t't''(right(t, t') \wedge down(t', t'') \rightarrow$$
$$\neg \exists t'''(down(t, t''') \wedge right(t''', t'')))$$
$$illegal() \leftarrow \exists t, t't''(down(t, t') \wedge right(t', t'') \rightarrow$$
$$\neg \exists t'''(right(t, t''') \wedge down(t''', t'')))$$

In addition, we will have axioms that verify that for all tiles there exist exactly the expected neighbors, e.g. that all non-corner tiles below the top left tile have no left tile but one up, right, and down.

**Goal** The PDDL goal of the Floortile tasks is a conjunction of *painted* atoms. In the domain specification, we use additional predicates $painted^g$ for encoding exactly these goal atoms already in the initial state in conjunction with the domain-wide goal $\forall t, c(painted^g(t, c) \rightarrow painted(t, c))$.

The axioms need to ensure that $painted^g$ defines a rectangular subset of the tiles painted with two colors like a chessboard. We leave the aspect of the rectangular subset to the reader, building on the ideas elaborated above for verifying the grid property. For the remaining properties, we need to ensure that the coloring uses exactly two colors, and that adjacent tiles in the rectangular area are colored differently.

$$gcolor(x) \leftarrow \exists t \, painted^g(t, x)$$
$$illegal() \leftarrow \neg \exists x, y(gcolor(x) \wedge gcolor(y) \wedge x \neq y)$$
$$illegal() \leftarrow \exists x, y, z(gcolor(x) \wedge gcolor(y) \wedge gcolor(z) \wedge$$
$$x \neq y \wedge x \neq z \wedge y \neq z)$$
$$illegal() \leftarrow \exists t, t', c(painted^g(t, c) \wedge painted^g(t', c) \wedge$$
$$(left(t, t') \vee down(t, t')))$$

## Grids Defined by Edge Relation

Many domains have a concept of an underlying grid graph that is represented by a binary predicate $E$ that represents the existence of an edge between two vertices. In contrast to the Floortile domain, there is no notion of cardinal directions in the edge predicates. One such example is the Grid domain from IPC 1998 (McDermott 2000), where an agent must use keys located on the grid to make further locations accessible to eventually reach a specific vertex on the grid. Other examples are Visitall from IPC 2011 (Coles et al. 2012), Termes from IPC 2018,[4] or Folding from IPC 2023.[5]

Here we focus on the question how we can use order support to verify that the interpretation of $E$ in the initial state corresponds to a grid graph. The encoding builds on the general idea we also used in Floortile. The crucial aspect is that we will use the order to select one corner as the top left one and one of the adjacent fringes as the top fringe.

We assume wlog. that all objects refer to vertices. Otherwise, we can identify them with a given predicate (e.g. *place* in the IPC Grid domain) or derive such a predicate from $E$. With this predicate it is then easy to restrict the following rules to vertices only. The following rules can identify all edge relations that represent an $n \times m$ grid with $m, n \geq 2$.

We verify that relation $E$ is symmetric and irreflexive:

$$illegal() \leftarrow \exists x, y(E(x, y) \wedge \neg E(y, x))$$
$$illegal() \leftarrow \exists x \, E(x, x)$$

Instead of the successor predicate *succ*, we will in the following use the corresponding linear order $<$ on the objects as predicate. This is always possible, because we can derive it from *succ* with the transitive closure:

$$x < y \leftarrow succ(x, y) \vee \exists z(x < z \wedge succ(z, y)).$$

We will exploit that a grid graph has corner vertices, which are the vertices with exactly two neighbors. We will use the smallest of them (with respect to the arbitrary order) to be used as the top left corner. This is always possible because the grid graph property is invariant under rotation. For this purpose, we introduce for $k \in \{1, \ldots, 5\}$ a predicate $n_{\geq k}(x)$ to express that vertex $x$ has at least $k$ neighbors.

$$n_{\geq k}(x) \leftarrow \exists x_1, \ldots, x_k \bigwedge_{i=1}^{k} \left( E(x, x_i) \wedge \bigwedge_{j=i+1}^{k} x_i \neq x_j \right)$$

---

[4]https://ipc2018-classical.bitbucket.io/
[5]https://ipc2023-classical.github.io/

A corner has exactly two neighbors and we mark the smallest corner as the top left corner:

$$corner(x) \leftarrow n_{\geq 2}(x) \wedge \neg n_{\geq 3}(x)$$
$$TL(x) \leftarrow corner(x) \wedge \neg \exists y(corner(y) \wedge y < x)$$

To help with the further steps, we use a predicate $betw(x, y, z)$ to express that vertex $x$ is directly between vertices $y$ and $z$ on a "straight line". In grid graphs this is the case iff the only path of length 2 from $y$ to $z$ goes through $x$.

$$betw(x, y, z) \leftarrow y \neq z \wedge \neg E(y, z) \wedge$$
$$E(x, y) \wedge E(x, z) \wedge$$
$$\forall x' (E(x', y) \wedge E(x', z) \rightarrow x = x')$$

As in Floortile, we want to associate each vertex with a row and column, using the vertices on the top and left fringe to identify columns and rows. We thus need to identify the other nodes on these fringes. As the grid property is invariant to mirroring on the diagonals, we can select the top fringe to be the one with the smaller neighbor of the top left vertex.

$$nextCol(x, y) \leftarrow \exists y'(TL(x) \wedge E(x, y) \wedge E(x, y') \wedge y < y')$$
$$nextCol(x, y) \leftarrow \exists z(betw(x, z, y) \wedge nextCol(z, x))$$
$$col(x) \leftarrow TL(x) \vee \exists y\, nextCol(y, x)$$
$$nextRow(x, y) \leftarrow \exists y'(TL(x) \wedge E(x, y) \wedge E(x, y') \wedge y' < y)$$
$$nextRow(x, y) \leftarrow \exists z(betw(x, z, y) \wedge nextRow(z, x))$$
$$row(x) \leftarrow TL(x) \vee \exists y\, nextRow(y, x)$$

Predicate $colOf(x, y)$ again represents that vertex $x$ is in column $y$. For the vertices in the top fringe this is the vertex itself. In a grid, all these vertices $v$ have exactly one neighbor that does not represent a column itself, namely the vertex below $v$. We use this property to define the predicate for the columns of the vertices in the second row. The remaining rows can then be resolved by means of $betw$.

$$colOf(x, x) \leftarrow col(x)$$
$$colOf(x, c) \leftarrow col(c) \wedge E(x, c) \wedge \neg col(x)$$
$$colOf(x, c) \leftarrow \exists y, z(colOf(y, c) \wedge colOf(z, c) \wedge$$
$$betw(y, x, z))$$

Predicate $rowOf(x, y)$ is defined analogously. Predicates $right$ (and $down$) can now be derived from the coordinates:

$$right(x, y) \leftarrow \exists c, c', r(colOf(x, c) \wedge colOf(y, c') \wedge$$
$$rowOf(x, r) \wedge rowOf(y, r) \wedge$$
$$nextRow(r, r'))$$

Predicates $up$ and $left$ can be defined as the inverse of $down$ and $right$. We are now set to verify as in Floortile that the coordinate embedding is a bijection and that $up$, $left$, $down$, $right$ have the right properties wrt. the existence of neighbors and the commutativity of the directions.

**Square Grids** The tasks of the Grid domain from the IPC always have square $n \times n$ grids (for different values of $n$). We can extend our axioms to this additional requirement.

We use the insight that the grid is square iff for each vertex on the left fringe there is a vertex on the top fringe with the same distance to the top left vertex, and vice versa. We cannot represent the distances explicitly but we can define a predicate $eqDist(x, y)$ to express that fringe vertices $x$ and $y$ have equal distance to the top left corner.

$$eqDist(x, y) \leftarrow TL(x) \wedge TL(y)$$
$$eqDist(x, y) \leftarrow \exists v(TL(v) \wedge E(x, v) \wedge E(y, v))$$
$$eqDist(x, y) \leftarrow \exists v, v', w, w'(eqDist(v, w) \wedge$$
$$eqDist(v', w') \wedge$$
$$betw(v', v, x) \wedge betw(w', w, y))$$
$$illegal() \leftarrow \exists r(row(r) \wedge \neg \exists c(col(c) \wedge eqDist(r, c)))$$
$$illegal() \leftarrow \exists c(col(c) \wedge \neg \exists r(row(r) \wedge eqDist(r, c)))$$

## Solvability of Transport

For domains where deciding solvability is in P, the planning domain can require all tasks to be solvable. We showcase this for the Transport domain,[6] where the goal is to deliver packages from their initial location to some destination location. Locations are connected by a symmetric *road* network, which can be used by trucks. Each truck can load packages up to its capacity to transport them between locations. Different capacities are represented by objects together with a *capacity-predecessor* relation. Here we focus on the solvability aspect and assume that there are additional axioms e.g. for ensuring the symmetry of the *road* relation and that there is a predicate $at^g$ for the destination of each package.

There is never the need to transport a package with multiple trucks, so a task is solvable if for each package there is a truck with a non-zero capacity (a capacity with a predecessor) that can reach its initial and goal location.

$$reach(t, l) \leftarrow at(t, l) \vee \exists l'(reach(t, l') \wedge road(l', l))$$
$$good(t) \leftarrow truck(t) \wedge \exists c, c'(capacity(t, c') \wedge$$
$$capacity\text{-}predecessor(c, c'))$$
$$illegal() \leftarrow \exists p, l, l'(at(p, l) \wedge at^g(p, l') \wedge l \neq l' \wedge$$
$$\neg \exists t(good(t) \wedge reach(t, l) \wedge reach(t, l')))$$

## Conclusion

We built on insights from database theory to propose a formalism for characterizing planning domains, and demonstrated its applicability on a number of established benchmark domains. Srivastava, Immerman, and Zilberstein (2011) already used a similar formalism for generalized planning. In particular, they also require a joint goal formula and define the legal initial states by means of logic. In contrast to us, they use first-order logic with transitive closure, which is strictly less expressive than stratified Datalog$^{\neg}$ (Grädel and McColm 1996) and thus also PDDL axioms.

In future work we plan to explore some of the possibilities mentioned in the introduction, starting with instance generation. In most domains, it is highly unlikely that a randomly guessed interpretation of the predicates corresponds to a legal initial state. We will therefore look into the exploitation of SAT solvers or answer set programming for this purpose.

---

[6]https://ipc08.icaps-conference.org/deterministic/HomePage.html

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
