# OpenReview forum: "Formal Representations of Classical Planning Domains"
_icaps-conference.org/ICAPS/2024/Conference — ICAPS 2024_

### Official Review · Reviewer_dKZm · 2024-01-22

**Significance And Importance:** 2
**Soundness:** 3
**Novelty:** 2
**Clarity:** 2
**Confidence:** 4

**Weaknesses:**

-1: Major weaknesses requiring significant work to be addressed for the paper to be accepted.

**Contributions Of The Paper:**

The paper seeks to formalise and tighten up the idea of a domain model in the basic form of the PDDL
language - a  useful research direction.
Their angle is to utilise PDDL axioms, and the logical query language Datalog.

**Ethical Considerations:**

(1) Not Applicable: The paper does not have any ethical considerations to address

**Nomination For Best Paper:**

No

**Overall Evaluation:**

-1: (weak reject)

**Questions For Authors:**

- can you state clearly which version of the PDDL family the work applies to,
and whether it is worth or possible to extend it up, say as far as PDDL+ ?

-- In the introduction it would be good to make exactly what you mean by the word
properties in "A subsequent extension allows us to express all properties that
can be decided in polynomial time" .. I assume properties are not the same as goals,
so its worth making this clearer.

-- can you mention anything about future work - are you aiming to create a tool to
help engineer domain models?

-- There have been several formalisations of the Blocks world - why is your formulation more
useful .. in relation to constraining "legal tasks beyond the possibilities of a PDDL domain"
I think you should make this explicit.

**Reproducibility:**

3: Authors describe the implementation and domains in sufficient detail.

**Strengths Of The Paper:**

- The paper seems to introduce a useful way of formalising classical planning
domain models, and promises to help in the classification / analysis and
engineering of those models.

- It includes a formal treatment of state expansion using axioms, and techniques
for deciding which problems were solvable, in polynomial time.

- It contains a range of examples from IPCs in the last 3 pages of the paper.

- The treatment looked fairly rigorous, though I'm not yet convinced of its significance (see below).

**Weaknesses Of The Paper:**

--- There is little or no Related Work discussion. To understand the significance of what is
proposed we need such a section. For Example: there has been a seam of work (> 20 years)
along these lines which the paper does not seem to be aware of - the work done and tools created in the
area of knowledge engineering for AI planning (e.g. KEPS workshops at ICAPS).
For example, if my recollection is clear the planning description language used in the tool GIPO (which was
a winner in the first ICKEPS competition) included a formal characterization of all tasks (and all legal states)
in the domain, using an object-centred language. One feature of GIPO was that it could be used to
generate "random" tasks - as the author(s) mention here is a useful tool.
Also, Vaquero's ITSimple did a simple job. Both these systems had PDDL translators,
and attempted to capture the intuitive notion of the domain model in a formal way,
and the language behind GIPO contained the ability for the user to formulate state invariants.

--- The paper could be very much improved with a much more in-discussion of the
implications of the work - while the overall thrust of the work to me is
understood, throughout the paper I was asking myself the question
why are we going down this particular path; e.g. up to Thm2 the treatment seems to wander - I was not
sure which direction the formalisms were taking the reader.

-- The paper could be improved with some concrete
motivating examples before page 4's logistics domain model example. For example,
starting with one of the IPC examples and illustrating the theory with this,
would help clarity the treatment and help motivation. Also, I think the contribution needs to be made clearer in the Introduction - why should we read the  paper and who is it addressed to?

Minor terminological remark: The paper seems to fall into the trap of using the word "domain"
where it means model (or "precise domain description"). It seems to me to refer
to the abstract reality one is trying to capture as the "Domain" and
the concrete formalism one uses as the "Model"
is less confusing - as it is acknowledged that there are many ways of encoding the
same domain (e.g. domain = block's world, but there are a wide range of models used to describe this domain ).

---

> ### Author Rebuttal · Authors · 2024-01-26
>
> We thank you for your comments.
>
> PDDL version: We consider PDDL 2.2 (including full ADL + axioms) without numerical or temporal features. The formalism can be used for richer variants, up to PDDL+, but receives as input only the purely logical aspects of the problem, so could not be used to model numerical constraints that restrict the allowed range of numerical problem aspects. Extensions with numerical constraints are possible, but then require trade-offs between expressivity and complexity if we want to preserve efficiency. For example, for full PDDL+ it is already undecidable whether a given PDDL+ problem is solved by the empty plan (because the autonomous processes can simulate unbounded Turing machines).
>
> “properties”: A property is anything that is either true or false for a given PDDL problem. This includes anything that can be computed by an algorithm that takes a PDDL problem as an input and deterministically returns true or false. In our case, we want to capture whether a given PDDL problem is part of the domain we want to model or not.
>
> future work: planned next steps include 1) publishing an enhanced IPC benchmark suite (not covering all domains, but at least 10+) with fully formal models in the sense of the paper; 2) a verifier that takes a domain (in the sense of the paper) and a PDDL problem as input and checks if the problem is legal for the domain, and 3) a domain-independent problem generator that generates random legal problems for a given input domain. A side-effect of 1) useful for domain modelers would be a catalog of reusable concepts like connected graph, grid, bijective function, acyclic relation.
>
> blocksworld formalization: we only use blocksworld as an illustrating example. The more general point is not that we can exactly describe legal blocksworld problems, but that for any domain and any existing PDDL model of the domain, we can exactly describe the legal instances as long as a polynomial-time algorithm testing legality exists (without modifying the PDDL model).
>
> Regarding related work, we will extend our discussion. GIPO is designed with different goals in mind; it is a knowledge engineering tool more than a formal specification tool. Its state-machine-based internal representations can only express (higher-order) mutex-style restrictions, which is less expressive than SAT-based representations, which are themselves too limited for concepts such as connectedness, as described in the answer to Reviewer Rxqs.

---

### Official Review · Reviewer_Rxqs · 2024-01-23

**Significance And Importance:** 1
**Soundness:** 4
**Novelty:** 2
**Clarity:** 4
**Overall Evaluation:** 1
**Confidence:** 3

**Weaknesses:**

1: Minor weaknesses that are easily fixable.

**Contributions Of The Paper:**

- Development of a formalism using PDDL axioms for the exact characterization of classical planning domains.
- Demonstration of the formalism’s applicability through examples from established IPC domains.
- Extension of the formalism to express properties decidable in polynomial time, enhancing its utility and applicability.

**Ethical Considerations:**

(1) Not Applicable: The paper does not have any ethical considerations to address

**Nomination For Best Paper:**

No

**Questions For Authors:**

1) What would be the computational complexity and scalability of the approach? Will it be practical?

2) Would you be able to provide a description in the paper of how to automate the encoding of PDDL domains in your formalism such that they can be used in practice?

3) What advantages does your method offer over existing logic-based methods in planning, which can be used to encode similar constraints? Can you clarify the novelty of your framework in comparison to that?

**Reproducibility:**

0: N/A - nothing to reproduce.

**Strengths Of The Paper:**

- Formalizing Planning Domains:
The paper's approach to using PDDL axioms for formalizing classical planning domains is noteworthy because it enhances the expressivity and flexibility of planning domain representations. Such an advancement is useful in addressing complex planning problems that standard representations might struggle to encapsulate effectively.

- Practical Demonstration Using Standard Examples:
The paper’s application of the formalism to well-known IPC domains demonstrates its practical utility.

- Theoretically Sound and Rigorous:
The use of PDDL axioms, coupled with Datalog¬, provides a robust and logically sound basis for the formalism.

- Explainability Potential:
The logic-based formalism has the potential of being used for explainability purposes.

**Weaknesses Of The Paper:**

- Applicability and Practicallity:
The paper does not address how the proposed formalism can be used in practice. For example, can it be used for answering unsolvability tasks? The authors state that it could, and that certain properties can be expressed in polynomial time, but it is not properly explored. So, what would be the computational complexity of an approach using the proposed formalism to answer unsolvability queries? Would such an approach scale?

- Lack of Discussion on Automation Potential:
The paper does not address the potential for automating the encoding of PDDL domains using the proposed formalism. This omission is a weakness because automation is a critical aspect of scalability and practical application. Addressing these is crucial if the formalism is to be used in practice (which I assume is the end-goal of the authors).

- Relevance to Logic-based Planning:
This may be a silly comment, but I don't really get the significance of the formalism compared to existing logic-based planning approaches, e.g., where a PDDL domain is encoded into a logical formalism (c.f. Kautz et al. '92)). From my point of view, we can encode all the legal constraints into a logical KB using a SAT-based formalism. Then, the KB will encode all these constraints and be as expressive as needed. It can also be used to answer specific queries and prove specific properties. So, what am I missing with my assessment?

---

> ### Author Rebuttal · Authors · 2024-01-26
>
> We thank you for your comments.
>
> Regarding Q1:
>
> The complexity properties of the formalism are the same as for grounding in the STRIPS fragment of PDDL, both for “data complexity” (fixed domain) and “query complexity” (allowing the domain to vary). With fixed domains (and thus with bounds on things like arity of predicates and number of action parameters), everything can be evaluated in polynomial time. The formalism we use is maximally expressive among those that can be evaluated in polynomial time because it can capture all polynomial-time computations (by any logic formalism or other computational paradigm). This is the key advantage over other formalisms like PDDL axioms without order or (lifted) SAT, which have the same computational properties, but are less expressive.
>
> Regarding Q2:
>
> Our aim is to use the proposed formalism to allow automating problems that cannot currently be fully automated because they cannot be formally stated (due to lack of comprehensive domain models), such as generating random benchmark tasks, or verifying that a generalized plan solves all tasks in the domain. The aim is not domain acquisition (creating the domain representation itself automatically). Nevertheless, once a formalism has been defined, automated domain acquisition in this formalism is a problem that could be studied, along the lines of work on inductive logic programming. This would extend the recent papers on domain model learning by Bonet, Geffner and collaborators (e.g., Rodriguez et al., KR 2021).
>
> Regarding Q3:
>
> SAT-based formalisms like Kautz and Selman’s (1992) can be viewed as special cases of the formalism we propose. Their main limitation is that they cannot model a notion of recursion or transitive closure, which means that they cannot express requirements like “In a legal blocks world state, there must not be a cycle in the ‘on’ relation.” (Indeed, Kautz and Selman discuss this in Section 3 “Anomalous Models”: their models can say things that are true of all intended models, but will in general also capture some unintended models.) This is not a question of just coming up with the right formula but a known fundamental limitation of predicate logic, first proved by Fagin in 1974. This also means that properties such as reachability or connectedness cannot be expressed as first-order logic formulas, whereas the formalism we propose can express all properties that can be computed in polynomial time.

---

### Official Review · Reviewer_TWqJ · 2024-01-24

**Significance And Importance:** 2
**Soundness:** 4
**Novelty:** 3
**Clarity:** 4
**Overall Evaluation:** 2
**Confidence:** 4

**Weaknesses:**

2: No major or minor weaknesses.

**Contributions Of The Paper:**

The paper focuses on the problem of providing a precise characterisation of the tasks to be associated with a domain. In general, PDDL domain descriptions provide an over-approximation of the legal tasks, as they admit tasks that should not be admitted. Resorting to a finite sets of tasks only is not a satisfactory way out.  To solve the problems, the authors propose a formalism based on PDDL axioms, based on a suitable fragment of Datalog, to be used to characterise the set of intended tasks (in general, an infinite number) of a domain. Such an ability can be exploited in a variety of contexts that are pointed out by the authors in the introduction. A minor extension of the proposed formalism is shown to allow one to specify any property that can be tested in polynomial time.

**Ethical Considerations:**

(1) Not Applicable: The paper does not have any ethical considerations to address

**Nomination For Best Paper:**

No

**Questions For Authors:**

The remark after Definition 5 is not completely clear to me. Can you better explain it?

**Reproducibility:**

0: N/A - nothing to reproduce.

**Strengths Of The Paper:**

The work builds on previous work by Thiébaux et al. on PDDL axioms as well as from notions and results from database theory. The contributions of the work are described in a clear and precise way. From a technical point of view, they are not surprising and particularly complex. Nevertheless, they are motivated and explained in a convincing way.  The effectiveness of the proposal is demonstrated on a number of benchmark domains.

The quality of the presentation is globally satisfactory and the paper is in general well written. I found the section about background both concise and informative, that is, at the right level of detail. The section on goal specification is also useful to clarify a number of issues that came to my mind while reading the previous sections. There are some minor things to fix here and there, e.g., to replace “e.g.” by “e.g.,”.

**Weaknesses Of The Paper:**

My only reservation on this work is about its actual impact on the areas mentioned in the introduction. A convincing application of the achieved results in one of these areas would make the proposal stronger and make its impact evident.

Minor points

Line 122: replace “basic state” by “a basic state”.

Page 1, pseudocode of Algorithm 1: if necessary, it can be safely omitted, as it does not add that much to the textual explanation.

Line 183: replace “but at the same time” by “, but, at the same time,”.

Line 186: replace “or” by “, or”.

Line 191: replace “but” by “, but” (the same at line 232).

Line 223: replace “with” by “, with”.

Line 242: replace “we assume” by “, we assume”.

Line 247: replace “we can” by “, we can”.

Line 301: replace “it is” by “, it is”.

Line 302: replace “but instead” by “, but, instead,”.

Lines 304-317. the example is not completely clear to me. Please, check whether everything is ok.

Line 321: replace “Then” by “Then,”.

Line 329: replace “as an example” by “, as an example,”.

Lines 344, 346, 348, and 393: replace “but” by “, but”.

Line 362: replace “then” by “, then”.

Line 371: replace “Then” by “then,”.

Line 399: replace “but” by “, but”.

Line 411: replace “it holds” by “, it holds”.

Line 413: replace “then” by “, then”.

Line 417: replace “but” by “, but”.

Line 483: replace “we use” by “, we use”.

Line 506: replace ”we will” by “, we will”.

Line 507: replace “but” by “, but” (the same in footnote 3).

Line 513: replace “i. e.” by “i.e.,” (the same at line 516).

Line 515: replace “blocks” by “, blocks”.

Line 519: replace “no block” by “, no block”.

Line 536: replace “and” by “, and” (the same at line 537).

Line 545: replace “Initially” by “Initially,”.

Line 570: replace “Instead” by “Instead,”.

Line 630: replace “predicate” by “predicate,”.

Line 708: replace “work” by “work,”.

---

> ### Author Rebuttal · Authors · 2024-01-26
>
> We thank you for your comments and in particular for the detailed list of minor points.
>
> Regarding your question, the word “query” is not used consistently in different papers in the related work, so we wanted to clarify which meaning we mean. The remark may indeed add rather than reduce confusion. We wanted to allude to the different meanings of “query” and informally rephrase the formal definition: that a query is a function mapping basic states (over a given set of constants) to “true” or “false”. It is probably a good idea to drop the allusion.

---

### Meta-Review · Area_Chair_qRKi · 2024-02-01

**Recommendation:** Accept (Poster)
**Confidence:** 3

**Metareview:**

This paper proposes a novel method for exactly characterising the planning tasks to be associated with a domain, based on axioms. The authors then demonstrate their method on several IPC domains.

The new theory and tools of the paper are interesting, novel, and relevant to the community.

The paper however still has noticeable problem when it comes to its quality of writing and scholarship. Especially previous knowledge-engineering methods (e.g. from KEPS) to describe planning domains should be discussed. (Note that even if their objective might be different than that of the authors, the spirit and ideas are related and should be discussed). In addition to the examples provided in the reviews, you should also consider the use of formal specification languages (like Z or B) from software engineering to capture the domains. Consider the work of Ammar, 2021 on Event-B and of West et al., 2002, on B, B-AMN, B-Tool trying to capture AI planning domains.

If the paper is ultimately accepted, I very strongly advise the authors to put an effort into this paper in order to improve its text related to general motivation, related work, and discussion and future work.

**Ethical Considerations:**

(1) Not Applicable: The paper does not have any ethical considerations to address